# Gradients of Generative Models for Improved Discriminative Analysis of Tandem Mass Spectra

**John T. Halloran**
Department of Public Health Sciences
University of California, Davis
jthalloran@ucdavis.edu

**David M. Rocke**
Department of Public Health Sciences
University of California, Davis
dmrocke@ucdavis.edu

## Abstract

Tandem mass spectrometry (*MS/MS*) is a high-throughput technology used to identify the proteins in a complex biological sample, such as a drop of blood. A collection of spectra is generated at the output of the process, each spectrum of which is representative of a peptide (protein subsequence) present in the original complex sample. In this work, we leverage the log-likelihood gradients of generative models to improve the identification of such spectra. In particular, we show that the gradient of a recently proposed dynamic Bayesian network (DBN) [7] may be naturally employed by a kernel-based discriminative classifier. The resulting Fisher kernel substantially improves upon recent attempts to combine generative and discriminative models for post-processing analysis, outperforming all other methods on the evaluated datasets. We extend the improved accuracy offered by the Fisher kernel framework to other search algorithms by introducing Theseus, a DBN representing a large number of widely used MS/MS scoring functions. Furthermore, with gradient ascent and max-product inference at hand, we use Theseus to learn model parameters without any supervision.

## 1 Introduction

In the past two decades, tandem mass spectrometry (*MS/MS*) has become an indispensable tool for identifying the proteins present in a complex biological sample. At the output of a typical MS/MS experiment, a collection of spectra is produced on the order of tens-to-hundreds of thousands, each of which is representative of a protein subsequence, called a *peptide*, present in the original complex sample. The main challenge in MS/MS is accurately identifying the peptides responsible for generating each output spectrum.

The most accurate identification methods search a database of peptides to first score peptides, then rank and return the top-ranking such peptide. The pair consisting of a scored candidate peptide and observed spectrum is typically referred to as a *peptide-spectrum match* (PSM). However, PSM scores returned by such database-search methods are often difficult to compare across different spectra (i.e., they are poorly calibrated), limiting the number of spectra identified per search [15]. To combat such poor calibration, post-processors are typically used to recalibrate PSM scores [13, 19, 20].

Recent work has attempted to exploit generative scoring functions for use with discriminative classifiers to better recalibrate PSM scores; by parsing a DBN's *Viterbi path* (i.e., the most probable sequence of random variables), heuristically derived features were shown to improve discriminative recalibration using support vector machines (SVMs). Rather than relying on heuristics, we look towards the more principled approach of a Fisher kernel [11]. Fisher kernels allow one to exploit the sequential-modeling strengths of generative models such as DBNs, which offer vast design flexibility for representing data sequences of varying length, for use with discriminative classifiers such as SVMs, which offer superior accuracy but often require feature vectors of fixed length. Although

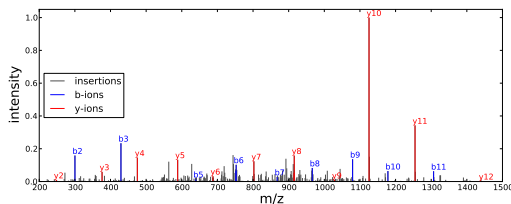

**Figure 1:** Example tandem mass spectrum with precursor charge $c(s) = 2$ and generating peptide $x =$ LWEPLLDVLVQTK. Plotted in red and blue are, respectively, b- and y-ion peaks (discussed in Section 2.1.1), while spurious observed peaks (called *insertions*) are colored gray. Note $y_1, b_1, b_4,$ and $b_{12}$ are absent fragment ions (called *deletions*).

the number of variables in a DBN may vary given different observed sequences, a Fisher kernel utilizes the fixed-length gradient of the log-likelihood (i.e., the *Fisher score*) in the feature-space of a kernel-based classifier. Deriving the Fisher scores of a DBN for Rapid Identification of Peptides (DRIP) [7], we show that the DRIP Fisher kernel greatly improves upon the previous heuristic approach; at a strict FDR of $1\%$ for the presented datasets, the heuristically derived DRIP features improve accuracy over the base feature set by an average $6.1\%$, while the DRIP Fisher kernel raises this average improvement to $11.7\%$ (Table 2 in [9]), thus nearly doubling the total accuracy of DRIP post-processing.

Motivated by improvements offered by the DRIP Fisher kernel, we look to extend this to other models by defining a generative model representative of the large class of existing scoring functions [2, 5, 6, 16, 10, 22, 17]. In particular, we define a DBN (called *Theseus*[1]) which, given an observed spectrum, evaluates the universe of all possible PSM scores. In this work, we use Theseus to model PSM score distributions with respect to the widely used XCorr scoring function [5]. The resulting Fisher kernel once again improves discriminative post-processing accuracy. Furthermore, with the generative model in place, we explore inferring parameters of the modeled scoring function using max-product inference and gradient-based learning. The resulting coordinate ascent learning algorithm outperforms standard maximum-likelihood learning. Most importantly, this overall learning algorithm is unsupervised which, to the authors' knowledge, is the first MS/MS scoring function parameter estimation procedure not to rely on any supervision. We note that this overall training procedure may be adapted by the many MS/MS search algorithms whose scoring functions lie in the class modeled by Theseus.

The paper is organized as follows. We discuss background information in Section 2, including the process by which MS/MS spectra are produced, the means by which spectra are identified, and related previous work. In Section 3, we extensively discuss the log-likelihood of the DRIP model and derive its Fisher scores. In Section 4, we introduce Theseus and derive gradients of its log-likelihood. We then discuss gradient-based unsupervised learning of Theseus parameters and present an efficient, monotonically convergent coordinate ascent algorithm. Finally, in Section 5, we show that DRIP and Theseus Fisher kernels substantially improve spectrum identification accuracy and that Theseus' coordinate ascent learning algorithm provides accurate unsupervised parameter estimation. For convenience, a table of the notation used in this paper may be found in [9].

## 2 Background

A typical tandem mass spectrometry experiment begins by cleaving proteins into peptides using a digesting enzyme. The resulting peptides are then separated via liquid chromatography and subjected to two rounds of mass spectrometry. The first round measures the mass and charge of the intact peptide, called the *precursor mass* and *precursor charge*, respectively. Peptides are then fragmented and the fragments undergo a second round of mass spectrometry, the output of which is an observed spectrum indicative of the fragmented peptide. The x-axis of this observed spectrum denotes *mass-to-charge* (*m/z*), measured in thomsons (Th), and the y-axis is a unitless intensity measure, roughly proportional to the abundance of a single fragment ion with a given m/z value. A sample such observed spectrum is illustrated in Figure 1.

## 2.1 MS/MS Database Search

Let $s$ be an observed spectrum with precursor mass $m(s)$ and precursor charge $c(s)$. In order to identify $s$, we search a database of peptides, as follows. Let $\mathcal{P}$ be the set of all possible peptide sequences. Each peptide $x \in \mathcal{P}$ is a string $x = x_1 x_2 \ldots x_n$ comprised of characters, called *amino acids*. Given a peptide database $\mathcal{D} \subseteq \mathcal{P}$, we wish to find the peptide $x \in \mathcal{D}$ responsible for generating $s$. Using the precursor mass and charge, the set of peptides to be scored is constrained by setting a mass tolerance threshold, $w$, such that we score the set of *candidate peptides* $D(s, \mathcal{D}, w) = \left\{ x : x \in \mathcal{D}, \left| \frac{m(x)}{c(s)} - m(s) \right| \leq w \right\}$, where $m(x)$ denotes the mass of peptide $x$. Note that we've overloaded $m(\cdot)$ to return either a peptide's or observed spectrum's precursor mass; we similarly overload $c(\cdot)$. Given $s$ and denoting an arbitrary scoring function as $\psi(x, s)$, the output of a search algorithm is thus $x^* = \operatorname{argmax}_{x \in D(m(s), c(s), \mathcal{D}, w)} \psi(x, s)$, the top-scoring PSM.

### 2.1.1 Theoretical Spectra

In order to score a candidate peptide $x$, fragment ions corresponding to suffix masses (called *b-ions*) and prefix masses (called *y-ions*) are collected into a *theoretical spectrum*. The annotated b- and y-ions of the generating peptide for an observed spectrum are illustrated in Figure 1. Varying based on the value of $c(s)$, the $k$th respective b- and y-ion pair of $x$ are

$$b(x, c_b, k) = \frac{\sum_{i=1}^{k} m(x_i) + c_b}{c_b}, \quad y(x, c_y, k) = \frac{\sum_{i=n-k}^{n} m(x_i) + 18 + c_y}{c_y},$$

where $c_b$ is the charge of the b-ion and $c_y$ is the charge of the y-ion. For $c(s) = 1$, we have $c_b = c_y = 1$, since these are the only possible, detectable fragment ions. For higher observed charge states $c(s) \geq 2$, it is unlikely for a single fragment ion to consume the entire charge, so that we have $c_b + c_y = c(s)$, where $c_b, c_y \in [1, c(s) - 1]$. The b-ion offset corresponds to the mass of a $c_b$ charged hydrogen atom, while the y-ion offset corresponds to the mass of a water molecule plus a $c_y$ charged hydrogen atom.

Further fragment ions may occur, each corresponding to the loss of a molecular group off a b- or y-ion. Called *neutral losses*, these correspond to a loss of either water, ammonia, or carbon monoxide. These fragment ions are commonly collected into a vector $v$, whose elements are weighted based on their corresponding fragment ion. For instance, XCorr [5] assigns all b- and y-ions a weight of 50 and all neutral losses a weight of 10.

## 2.2 Previous Work

Many scoring functions have been proposed for use in search algorithms. They range from simple dot-product scoring functions (X!Tandem [2], Morpheus [22]), to cross-correlation based scoring functions (XCorr [5]), to exact $p$-values over linear scoring functions calculated using dynamic programming (MS-GF+ [16] and XCorr $p$-values [10]). The recently introduced DRIP [7] scores candidate peptides without quantization of m/z measurements and allows learning the expected locations of theoretical peaks given high quality, labeled training data. In order to avoid quantization of the m/z axis, DRIP employs a dynamic alignment strategy wherein two types of prevalent phenomena are explicitly modeled: spurious observed peaks, called *insertions*, and absent theoretical peaks, called *deletions* (examples of both are displayed in Figure 1). DRIP then uses max-product inference to calculate the most probable sequences of insertions and deletions to score candidate peptides, and was shown to achieve state-of-the-art performance on a variety of datasets.

In practice, scoring functions are often *poorly calibrated* (i.e., PSM scores from different spectra are difficult to compare to one another), leading to potentially identified spectra left on the table during statistical analysis. In order to properly recalibrate such PSM scores, several semi-supervised post-processing methods have been proposed [13, 19, 20]. The most popular such method is Percolator [13], which, given the output target and decoy PSMs (discussed in Section 5) of a scoring algorithm and features detailing each PSM, utilizes an SVM to learn a discriminative classifier between target PSMs and decoy PSMs. PSM scores are then recalibrated using the learned decision boundary.

Recent work has attempted to leverage the generative nature of the DRIP model for discriminative use by Percolator [8]. As earlier mentioned, the output of DRIP is the most probable sequence of insertions and deletions, i.e., the Viterbi path. However, DRIP's observations are the sequences of

observed spectrum m/z and intensity values, so that the lengths of PSM's Viterbi paths vary depending on the number of observed spectrum peaks. In order to exploit DRIP's output in the feature-space of a discriminative classifier, PSM Viterbi paths were heuristically mapped to a fixed-length vector of features. The resulting heuristic features were shown to dramatically improve Percolator's ability to discriminate between PSMs.

## 2.3 Fisher Kernels

Using generative models to extract features for discriminative classifiers has been used to great effect in many problem domains by using Fisher kernels [11, 12, 4]. Assuming a generative model with a set of parameters $\theta$ and likelihood $p(O|\theta) = \sum_H p(O, H|\theta)$, where $O$ is a sequence of observations and $H$ is the set of hidden variables, the *Fisher score* is then $U_o = \nabla_\theta \log p(O|\theta)$. Given observations $O_i$ and $O_j$ of differing length (and, thus, different underlying graphs in the case of dynamic graphical models), a kernel-based classifier over these instances is trained using $U_{O_i}$ and $U_{O_j}$ in the feature-space. Thus, a similarity measure is learned in the gradient space, under the intuition that objects which induce similar likelihoods will induce similar gradients.

## 3 DRIP Fisher Scores

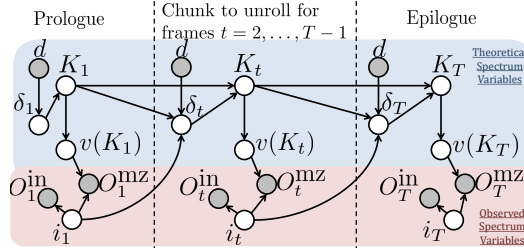

**Figure 2:** Graph of DRIP, the frames (i.e., time instances) of which correspond to observed spectrum peaks. Shaded nodes represent observed variables and unshaded nodes represent hidden variables. Given an observed spectrum, the middle frame (the chunk) dynamically expands to represent the second observed peak to the penultimate observed peak.

We first define, in detail, DRIP's log-likelihood, followed by the Fisher score derivation for DRIP's learned parameters. For discussion of the DRIP model outside the score of this work, readers are directed to [7, 8]. Denoting an observed peak as a pair $(O^{\mathrm{mz}}, O^{\mathrm{in}})$ consisting of an m/z measurement and intensity measurement, respectively, let $s = (O_1^{\mathrm{mz}}, O_1^{\mathrm{in}}), (O_2^{\mathrm{mz}}, O_2^{\mathrm{in}}), \ldots, (O_T^{\mathrm{mz}}, O_T^{\mathrm{in}})$ be an MS/MS spectrum of $T$ peaks and $x$ be a candidate (which, given $s$, we'd like to score). We denote the theoretical spectrum of $x$, consisting of its unique b- and y-ions sorted in ascending order, as the length-$d$ vector $v$. The graph of DRIP is displayed in Figure 2, where variables which control the traversal of the theoretical spectrum are highlighted in blue and variables which control the scoring of observed peak measurements are highlighted in red. Groups of variables are collected into time instances called *frames*. The frames of DRIP correspond to the observed peak m/z and intensity observations, so that there are $T$ frames in the model.

Unless otherwise specified, let $t$ be an arbitrary frame $1 \leq t \leq T$. $\delta_t$ is a multinomial random variable which dictates the number of theoretical peaks traversed in a frame. The random variable $K_t$, which denotes the index of the current theoretical peak index, is a deterministic function of its parents, such that $p(K_t = K_{t-1} + \delta_t | K_{t-1}, \delta_t) = 1$. Thus, $\delta_t > 1$ corresponds to the deletion of $\delta_t - 1$ theoretical peaks. The parents of $\delta_t$ ensure that DRIP does not attempt to increment past the last theoretical peak, i.e., $p(\delta_t > d - K_{t-1} | d, K_{t-1}, i_{t-1}) = 0$. Subsequently, the theoretical peak value $v(K_t)$ is used to access a Gaussian from a collection (the mean of each Gaussian corresponds to a position along the m/z axis, learned using the EM algorithm [3]) with which to score observations. Hence, the state-space of the model is all possible traversals, from left to right, of the theoretical spectrum, accounting for all possible deletions.

When scoring observed peak measurements, the Bernoulli random variable $i_t$ denotes whether a peak is scored using learned Gaussians (when $i_t = 0$) or considered an insertion and scored using an

insertion penalty (when $i_t = 1$). When scoring m/z observations, we thus have $p(O_t^{\text{mz}}|v(K_t), i_t = 0) = f(O_t^{\text{mz}}|\mu^{\text{mz}}(v(K_t)), \sigma^2)$ and $p(O_t^{\text{mz}}|v(K_t), i_t = 1) = a_{\text{mz}}$, where $\mu^{\text{mz}}$ is a vector of Gaussian means and $\sigma^2$ the m/z Gaussian variance. Similarly, when scoring intensity observations, we have $p(O_t^{\text{in}}|i_t = 0) = f(O_t^{\text{in}}|\mu^{\text{in}}, \bar{\sigma}^2)$ and $p(O_t^{\text{in}}|i_t = 1) = a_{\text{in}}$, where $\mu^{\text{in}}$ and $\bar{\sigma}^2$ are the intensity Gaussian mean and variance, respectively. Let $i_0 = K_0 = \emptyset$ and $\mathbf{1}_{\{.\}}$ denote the indicator function. Denoting DRIP's Gaussian parameters as $\theta$, the likelihood is thus

$$
\begin{aligned}
p(s|x,\theta) &= \prod_{t=1}^{T} p(\delta_t|K_{t-1}, d, i_{t-1})\mathbf{1}_{\{K_t = K_{t-1}+\delta_t\}} p(O_t^{\text{mz}}|K_t)p(O_t^{\text{in}}) \\
&= \prod_{t=1}^{T} p(\delta_t|K_{t-1}, d, i_{t-1})\mathbf{1}_{\{K_t = K_{t-1}+\delta_t\}} \left(\sum_{i_t=0}^{1} p(i_t)p(O_t^{\text{mz}}|K_t, i_t)\right)\left(\sum_{i_t=0}^{1} p(i_t)p(O_t^{\text{in}}|i_t)\right) \\
&= \prod_{t=1}^{T} \phi(\delta_t, K_{t-1}, i_t, i_{t-1}).
\end{aligned}
$$

The only stochastic variables in the model are $i_t$ and $\delta_t$, where all other random variables are either observed or deterministic given the sequences $i_{1:T}$ and $\delta_{1:T}$. Thus, we may equivalently write $p(s|x,\theta) = p(i_{1:T}, \delta_{1:T}|\theta)$. The Fisher score of the $k$th m/z mean is thus $\frac{\partial}{\partial \mu^{\text{mz}}(k)} \log p(s|x,\theta) = \frac{1}{p(s|x,\theta)} \frac{\partial}{\partial \mu^{\text{mz}}(k)} p(s|x,\theta)$, and we have (please see [9] for the full derivation)

$$
\begin{aligned}
\frac{\partial}{\partial \mu^{\text{mz}}(k)} p(s|x,\theta) &= \frac{\partial}{\partial \mu^{\text{mz}}(k)} \sum_{i_{1:T}, \delta_{1:T}} p(i_{1:T}, \delta_{1:T}|\theta) = \sum_{i_{1:T}, \delta_{1:T}: K_t = k, 1 \leq t \leq T} \frac{\partial}{\partial \mu^{\text{mz}}(k)} p(i_{1:T}, \delta_{1:T}|\theta) \\
&= \sum_{i_{1:T}, \delta_{1:T}} \mathbf{1}_{\{K_t=k\}} p(s|x,\theta) \left(\prod_{t:K_t=k} \frac{1}{p(O_t^{\text{mz}}|K_t)}\right)\left(\frac{\partial}{\partial \mu^{\text{mz}}(k)} \prod_{t:K_t=k} p(O_t^{\text{mz}}|K_t)\right).
\end{aligned}
$$

$$
\Rightarrow \frac{\partial}{\partial \mu^{\text{mz}}(k)} \log p(s|x,\theta) = \sum_{t=1}^{T} p(i_t, K_t = k|s,\theta)p(i_t = 0|K_t, O_t^{\text{mz}})\frac{(O_t^{\text{mz}} - \mu^{\text{mz}}(k))}{\sigma^2}. \quad (1)
$$

Note that the posterior in Equation 1, and thus the Fisher score, may be efficiently computed using sum-product inference. Through similar steps, we have

$$
\frac{\partial}{\partial \sigma^2(k)} \log p(s|x,\theta) = \sum_{t} p(i_t, K_t = k|s,\theta)p(i_t = 0|K_t, O_t^{\text{mz}})\left(\frac{(O_t^{\text{mz}} - \mu^{\text{mz}}(k))}{2\sigma^4} - \frac{1}{2\sigma^2}\right) \quad (2)
$$

$$
\frac{\partial}{\partial \mu^{\text{in}}} \log p(s|x,\theta) = \sum_{t} p(i_t, K_t|s,\theta)p(i_t = 0|O_t^{\text{in}})\frac{(O_t^{\text{in}} - \mu^{\text{in}})}{\bar{\sigma}^2} \quad (3)
$$

$$
\frac{\partial}{\partial \bar{\sigma}^2} \log p(s|x,\theta) = \sum_{t} p(i_t, K_t|s,\theta)p(i_t = 0|O_t^{\text{in}})\left(\frac{(O_t^{\text{in}} - \mu^{\text{in}})}{2\bar{\sigma}^4} - \frac{1}{2\bar{\sigma}^2}\right), \quad (4)
$$

where $\sigma^2(k)$ denotes the partial derivative of the variance for the $k$th m/z Gaussian with mean $\mu^{\text{mz}}(k)$.

Let $U_\mu = \nabla_{\mu^{\text{mz}}} \log p(s, x|\theta)$ and $U_{\sigma^2} = \nabla_{\sigma^2(k)} \log p(s, x|\theta)$. $U_\mu$ and $U_{\sigma^2}$ are length-$d$ vectors corresponding to the mapping of a peptide's sequence of b- and y-ions into $r$-dimensional space (i.e., dimension equal to an m/z-discretized observed spectrum). Let $\mathbb{1}$ be the length-$r$ vector of ones. Defining $z^{\text{mz}}, z^{\text{i}} \in \mathbb{R}^r$, the elements of which are the quantized observed spectrum m/z and intensity values, respectively, we use the following DRIP gradient-based features for SVM training in Section 5: $|U_\mu|_1, |U_{\sigma^2}|_1, U_\mu^T z^{\text{mz}}, U_{\sigma^2}^T z^{\text{i}}, U_\mu^T \mathbb{1}, U_{\sigma^2}^T \mathbb{1}, \frac{\partial}{\partial \mu^{\text{in}}} \log p(s, x|\theta)$, and $\frac{\partial}{\partial \bar{\sigma}^2} \log p(s, x|\theta)$.

## 4 Theseus

Given an observed spectrum $s$, we focus on representing the universe of linear PSM scores using a DBN. Let $z$ denote the vector resulting from preprocessing the observed spectrum, $s$. As a modeling example, we look to represent the popular XCorr scoring function. Using subscript $\tau$ to denote a

vector whose elements are shifted $\tau$ units, XCorr's scoring function is defined as

$$\mathrm{XCorr}(s, x) = v^T z - \sum_{\tau=-75}^{75} v^T z_\tau = v^T(z - \sum_{\tau=-75}^{75} z_\tau) = v^T z',$$

where $z' = z - \sum_{\tau=-75}^{75} z_\tau$. Let $\theta \in \mathbb{R}^l$ be a vector of XCorr weights for the various types of possible fragment ions (described in Section 2.1.1). As described in [10], given $c(s)$, we reparameterize $z'$ into a vector $z_\theta$ such that $\mathrm{XCorr}(x, s)$ is rendered as a dot-product between $z_\theta$ and a boolean vector $u$ in the reparameterized space. This reparameterization readily applies to any linear MS/MS scoring function. The $i$th element of $z_\theta$ is $z_\theta(i) = \sum_{j=1}^{l} \theta(j)z_j(i)$, where $z_j$ is a vector whose element $z_j(i)$ is the sum of all higher charged fragment ions added into the singly-charged fragment ions for the $j$th fragment ion type. The nonzero elements of $u$ correspond to the singly-charged b-ions of $x$ and we have $u^T z_\theta = \sum_{i=1}^{n} z_\theta(m(x_i) + 1) = \sum_{i=1}^{n} \sum_{j=1}^{l} \theta(j)z_j(m(x_i) + 1) = v^T z' = \mathrm{XCorr}(s, x)$.

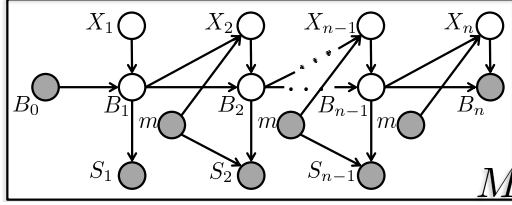

**Figure 3:** Graph of Theseus. Shaded nodes are observed random variables and unshaded nodes are hidden (i.e., stochastic). The model is unrolled for $n + 1$ frames, including $B_0$ in frame zero. Plate notation denotes $M$ repetitions of the model, where $M$ is the number of discrete precursor masses allowed by the precursor-mass tolerance threshold, $w$.

Our generative model is illustrated in Figure 3. $n$ is the maximum possible peptide length and $m$ is one of $M$ discrete precursor masses (dictated by the precursor-mass tolerance threshold, $w$, and $m(s)$). A *hypothesis* is an instantiation of random variables across all frames in the model, i.e., for the set of all possible sequences of $X_i$ random variables, $X_{1:n} = X_1, X_2, \ldots, X_n$, a hypothesis is $x_{1:n} \in X_{1:n}$. In our case, each hypothesis corresponds to a peptide and the corresponding log-likelihood its XCorr score. Each frame after the first contains an amino acid random variable so that we accumulate b-ions in successive frames and access the score contribution for each such ion.

For frame $i$, $X_i$ is a random amino acid and $B_i$ the accumulated mass up to the current frame. $B_0$ and $B_n$ are observed to zero and $m$, respectively, enforcing the boundary conditions that all length-$n$ PSMs considered begin with mass zero and end at a particular precursor mass. For $i > 0$, $B_i$ is a deterministic function of its parents, $p(B_i|B_{i-1}, X_i) = p(B_i = B_{i-1} + m(X_i)) = 1$. Thus, hypotheses which do not respect these mass constraints receive probability zero, i.e., $p(B_n \neq m|B_{n-1}, X_n) = 0$. $m$ is observed to the value of the current precursor mass being considered.

Let $\mathcal{A}$ be the set of amino acids, where $|\mathcal{A}| = 20$. Given $B_i$ and $m$, the conditional distribution of $X_i$ changes such that $p(X_i \in \mathcal{A}|B_{i-1} < m) = \alpha \mathcal{U}\{\mathcal{A}\}, p(X_i = \kappa|B_{i-1} \geq m) = 1$, where $\mathcal{U}\{\cdot\}$ is the uniform distribution over the input set and $\kappa \notin \mathcal{A}, m(\kappa) = 0$. Thus, when the accumulated mass is less than $m$, $X_i$ is a random amino acid and, otherwise, $X_i$ deterministically takes on a value with zero mass. To recreate XCorr scores, $\alpha = 1/|\mathcal{A}|$, though, in general, any desired mass function may be used for $p(X_i \in \mathcal{A}|B_{i-1} < m)$.

$S_i$ is a *virtual evidence child* [18], i.e., a leaf node whose conditional distribution need not be normalized to compute probabilistic quantities of interest in the DBN. For our model, we have $p(S_i|B_i < m, \theta) = \exp(z_\theta(B_i)) = \exp(\sum_{i=1}^{|\theta|} \theta_i z_i(B_i))$ and $p(S_i|B_i \geq m, \theta) = 1$. Let $t'$ denote

the first frame in which $m(X_{1:n}) \geq m$. The log-likelihood is then $\log p(s, X_{1:n}|\theta)$

$$= \log p(X_{1:n}, B_{0:n}, S_{1:n-1})$$

$$= \log(\mathbf{1}_{\{B_0=0\}}(\prod_{i=1}^{n-1} p(X_i|m, B_{i-1})p(B_i = B_{i-1} + m(X_i))p(S_i|m, B_i, \theta))\mathbf{1}_{\{B_{n-1}+m(X_n)=m\}})$$

$$= \log \mathbf{1}_{\{B_0=0 \,\wedge\, m(X_{1:n})=m\}} + \log(\prod_{i=t'+1}^{n} p(X_i|m, B_{i-1})p(B_i = B_{i-1} + m(X_i))p(S_i|m, B_i, \theta))+$$

$$\log(\prod_{i=1}^{t'} p(X_i|m, B_{i-1})p(B_i = B_{i-1} + m(X_i))p(S_i|m, B_i, \theta))$$

$$= \log \mathbf{1}_{\{m(X_{1:n})=m\}} + \log 1 + \log(\prod_{i=1}^{t'} \exp(z_\theta(B_i)))$$

$$= \log \mathbf{1}_{\{m(X_{1:n})=m\}} + \sum_{i=1}^{t'} z_\theta(B_i) = \log \mathbf{1}_{\{B_0=0 \,\wedge\, m(X_{1:n})=m\}} + \text{XCorr}(X_{1:t'}, s)$$

The $i$th element of Theseus' Fisher score is thus

$$\frac{\partial}{\partial \theta(i)} \log p(s|\theta) = \frac{\partial}{\partial \theta(i)} \log \sum_{x_{1:n}} p(s, x_{1:n}|\theta) = \frac{1}{p(s|\theta)} \frac{\partial}{\partial \theta(i)} \sum_{x_{1:n}} p(s, x_{1:n}|\theta)$$

$$= \frac{1}{p(s|\theta)} \sum_{x_{1:n}} \mathbf{1}_{\{b_0=0 \,\wedge\, m(x_{1:n})=m\}} (\sum_{j=1}^{t'} z_i(b_j)) \prod_{j=1}^{t'} \exp(z_\theta(b_j)), \quad (5)$$

While Equation 5 is generally difficult to compute, we calculate it efficiently using sum-product inference. Note that when the peptide sequence is observed, i.e., $X_{1:n} = \hat{x}$, we have $\frac{\partial}{\partial \theta(i)} \log p(s, \hat{x}|\theta) = \sum_j z(m(\hat{x}_{1:j}))$.

Using the model's Fisher scores, Theseus' parameters $\theta$ may be learned via maximum likelihood estimation. Given a dataset of spectra $s^1, s^2, \ldots, s^n$, we present an alternate learning algorithm (Algorithm 1) which converges monotonically to a local optimum (proven in [9]). Within each iteration, Algorithm 1 uses max-product inference to efficiently infer the most probable PSMs per iteration, mitigating the need for training labels. $\theta$ is maximized in each iteration using gradient ascent.

---

**Algorithm 1** Theseus Unsupervised Learning Algorithm
___
1: **while** not converged **do**
2:     **for** $i = 1, \ldots, n$ **do**
3:         $\hat{x}^i \leftarrow \text{argmax}_{x^i \in \mathcal{P}} \log p(s^i, x^i|\theta)$
4:     **end for**
5:     $\theta \leftarrow \text{argmax}_\theta \sum_{i=1}^n \log p(s^i, \hat{x}^i|\theta)$
6: **end while**
___

## 5 Results

Measuring peptide identification performance is complicated by the fact that ground-truth is unavailable for real-world data. Thus, in practice, it is most common to estimate the *false discovery rate* (*FDR*) [1] by searching a decoy database of peptides which are unlikely to occur in nature, typically generated by shuffling entries in the target database [14]. For a particular score threshold, $t$, FDR is then calculated as the proportion of decoys scoring better than $t$ to the number of targets scoring better than $t$. Once the target and decoy PSMs are calculated, a curve displaying the FDR threshold vs. the number of correctly identified targets at each given threshold may be calculated. In place of FDR along the x-axis, we use the *q-value* [14], defined to be the minimum FDR threshold at which a given score is deemed to be significant. As many applications require a search algorithm perform well at low thresholds, we only plot $q \in [0, 0.1]$.

The same datasets and search settings used to evaluate DRIP's heuristically derived features in [8] are adapted in this work. MS-GF+ (one of the most accurate search algorithms in wide use, plotted

for reference) was run using version 9980, with PSMs ranked by E-value and Percolator features calculated using `msgf2pin`. All database searches were run using a $\pm 3.0$Th mass tolerance, XCorr flanking peaks not allowed in Crux searches, and all search algorithm settings otherwise left to their defaults. Peptides were derived from the protein databases using trypsin cleavage rules without suppression of proline and a single fixed carbamidomethyl modification was included.

Gradient-based feature representations derived from DRIP and XCorr were used to train an SVM classifier [13] and recalibrate PSM scores. Theseus training and computation of XCorr Fisher scores were performed using a customized version of Crux v2.1.17060 [17]. For an XCorr PSM, a feature representation is derived directly using both $\nabla_\theta \log p(s|\theta)$ and $\nabla_\theta \log p(s, x|\theta)$ as defined in Section 4, representing gradient information for both the distribution of PSM scores and the individual PSM score, respectively. DRIP gradient-based features, as defined in Section 3, were derived using a customized version of the DRIP Toolkit [8].Figure 4 displays the resulting search accuracy for four worm and yeast datasets. For the uncalibrated search results in Figure 5, we show that XCorr parameters may be learned without supervision using Theseus, and that the presented coordinate descent algorithm (which estimates the most probable PSMs to take a step in the objective space) converges to a much better local optimum than maximum likelihood estimation.

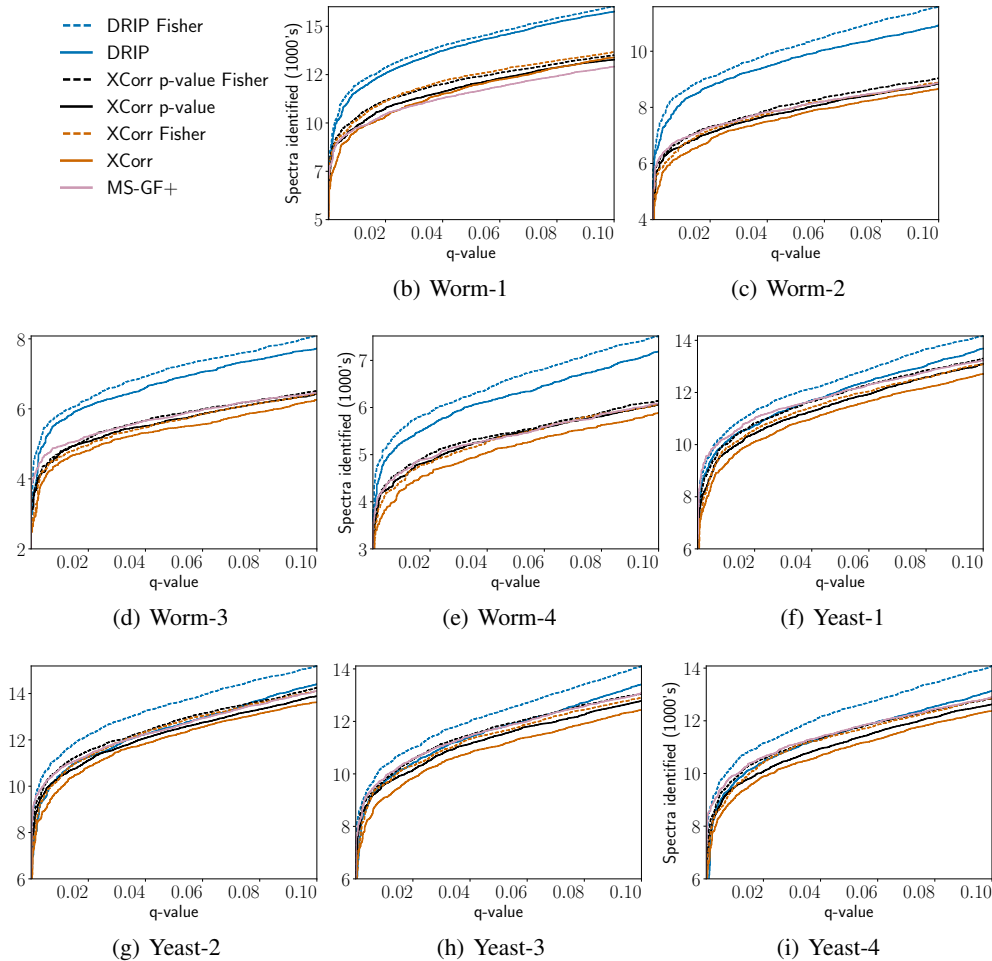

**Figure 4:** Search accuracy plots measured by $q$-value versus number of spectra identified for worm (*C. elegans*) and yeast (*Saccharomyces cerevisiae*) datasets. All methods are post-processed using the Percolator SVM classifier [13]. "DRIP" augments the standard set of DRIP features with DRIP-Viterbi-path parsed PSM features (described in [8]) and "DRIP Fisher" augments the heuristic set with gradient-based DRIP features. "XCorr," "XCorr $p$-value," and "MS-GF+" use their standard sets of Percolator features (described in [8]), while "XCorr $p$-value Fisher" and "XCorr Fisher" augment the standard XCorr feature sets with gradient-based Theseus features.

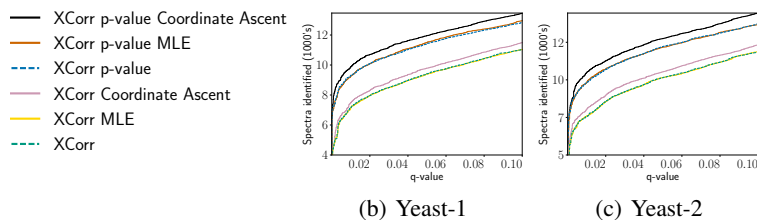

(b) Yeast-1      (c) Yeast-2

**Figure 5:** Search accuracy of Theseus' learned scoring function parameters. Coordinate ascent parameters are learned using Algorithm 1 and MLE parameters are learned using gradient ascent.

## 5.1 Discussion

DRIP gradient-based post-processing improves upon the heuristically derived features in all cases, and does so substantially on a majority of datasets. In the case of the yeast datasets, this distinguishes DRIP post-processing performance from all competitors and leads to state-of-the-art identification accuracy. Furthermore, we note that both XCorr and XCorr $p$-value post-processing performance are greatly improved using the gradient-based features derived using Theseus, raising performance above the highly similar MS-GF+ in several cases. Particularly noteworthy is the substantial improvement in XCorr accuracy which, using gradient-based information, is nearly competitive with its $p$-value counterpart. Considering the respective runtimes of the underlying search algorithms, this thus presents a tradeoff for a researcher considering search time and accuracy. In practice, the DRIP and XCorr $p$-value computations are at least an order of magnitude slower than XCorr computation in Crux [21]. Thus, the presented work not only improves state-of-the-art accuracy, but also improves the accuracy of simpler, yet significantly faster, search algorithms.

Owing to max-product inference in graphical models, we also show that Theseus may be used to effectively learn XCorr model parameters (Figure 5) without supervision. Furthermore, we show that XCorr $p$-values are also made more accurate by training the underlying scoring function for which $p$-values are computed. This marks a novel step towards unsupervised training of uncalibrated scoring functions, as unsupervised learning has been extensively explored for post-processor recalibration, but has remained an open problem for MS/MS database-search scoring functions. The presented learning framework, as well as the presented XCorr gradient-based feature representation, may be adapted by many of the widely scoring functions represented by Theseus [2, 5, 6, 16, 10, 22, 17].

Many exciting avenues are open for future work. Leveraging the large breadth of graphical models research, we plan to explore other learning paradigms using Theseus (for instance, estimating other PSMs using $k$-best Viterbi in order to discriminatively learn parameters using algorithms such as max-margin learning). Perhaps most exciting, we plan to further investigate the peptide-to-observed-spectrum mapping derived from DRIP Fisher scores. Under this mapping, we plan to explore learning distance metrics between PSMs in order to identify proteins from peptides.

**Acknowledgments**: This work was supported by the National Center for Advancing Translational Sciences (NCATS), National Institutes of Health, through grant UL1 TR001860.

## Footnotes

[1]In honor of Shannon's magnetic mouse, which could learn to traverse a small maze.

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
