[Supplementary Material]

# Gradients of Generative Models for Improved Discriminative Analysis of Tandem Mass Spectra: Supplementary Materials

**John T. Halloran**
Department of Public Health Sciences
University of California, Davis
jthalloran@ucdavis.edu

**David M. Rocke**
Department of Public Health Sciences
University of California, Davis
dmrocke@ucdavis.edu

## 1 Notation

| Symbol | Description |
| --- | --- |
| $s$ | observed spectrum of length $T$ |
| $m(s)$ | observed spectrum precursor mass |
| $c(s)$ | observed spectrum precursor charge |
| $O^{\text{mz}}$ | Observed peak m/z measurement |
| $O^{\text{in}}$ | Observed peak intensity measurement |
| $\mathcal{P}$ | set of all possible peptides |
| $x$ | peptide string, of length $n$, comprised of amino acids, $x = x_1, x_2, \ldots, x_n = x_{1:n}$ |
| $c(x)$ | charge of peptide $x$ (note the overloaded use of the charge operator) |
| $m(x_{1:t})$ | mass of length-$t$ peptide $x_{1:t}$ (note the overloaded use of the mass operator) |
| $\mathcal{D}$ | database of peptides to be searched |
| $w$ | mass tolerance threshold, used to filter peptides during search |
| $D(s, \mathcal{D}, w)$ | candidate set of peptides to be scored and ranked in order to identify $s$ |
| $c_b$ and $c_y$ | b-ion and y-ion charge, respectively, such that $c_b = c_y = 1$ for $c(s) = 1$ and $c_b + c_y = c(s)$ for $c(s) > 1$. |
| $b(x, c_b, k), y(x, c_y, k)$ | $k$th b- and y-ion pair of $x$ |
| $v$ | length-$d$ theoretical spectrum of $x$ |
| $t$ | arbitrary frame value for DRIP |
| $\delta_t$ | DRIP random variable signifying the number of theoretical peaks to move down in frame $t$; $\delta_t > 1$ corresponds to a deletion event |
| $i_t$ | DRIP Bernoulli random variable signifying whether an observed peak is scored as an insertion or not |
| $K_t$ | DRIP random variable signifying the theoretical peak index in frame $t$ |
| $a_{\text{mz}}$ | DRIP m/z insertion penalty |
| $a_{\text{in}}$ | DRIP intensity insertion penalty |
| $\mu^{\text{mz}}$ | vector of DRIP's m/z Gaussian means |
| $\sigma^2$ | DRIP's m/z Gaussian variance |
| $\mu^{\text{in}}$ | DRIP's intensity Gaussian mean |
| $\bar{\sigma}^2$ | DRIP's intensity Gaussian variance |
| $\theta$ | generative model's learnable parameters: for DRIP, this corresponds to all Gaussian means and variances; for Theseus and the modeled XCorr scoring function, this corresponds to the fragment ion weights |
| $\tau$ | XCorr vector shift increment |
| $z$ | vector resulting from (XCorr) quantization and preprocessing of $s$ |
| $z'$ | final step of XCorr preprocessing, where $z' = z - \sum_{\tau=-75}^{75} z_\tau$ |

| | | |
|---|---|---|
| $z_\theta$ | reparameterized observed spectrum vector, such that a linear score may be computed as the product of $z_\theta$ and a boolean theoretical vector $u$ | |
| $u$ | boolean theoretical vector with nonzero entries corresponding to the unity charge b-ions of $x$ | |
| $M$ | set of discrete precursor masses, dictated by $w$, iterated over in Theseus | |
| $X_{1:n}$ | random peptide modeled in Theseus | |
| $B_i$ | Theseus accumulated mass up to frame $i$ | |
| $\mathcal{A}$ | set of amino acids | |
| $S_i$ | Theseus virtual evidence child in frame $i$ | |

**Table 1:** Notation used in the main paper.

## 2 DRIP Fisher Score Derivation

Following the discussion in Section 3 of [2], $\frac{\partial}{\partial \mu^{\text{mz}}(k)} \log p(s|x,\theta) = \frac{1}{p(s|x,\theta)} \frac{\partial}{\partial \mu^{\text{mz}}(k)} p(s|x,\theta)$ and we have $\frac{\partial}{\partial \mu^{\text{mz}}(k)} p(s|x,\theta)$

$$= \frac{\partial}{\partial \mu^{\text{mz}}(k)} \sum_{i_{1:T},\delta_{1:T}} p(i_{1:T},\delta_{1:T}|\theta) = \sum_{i_{1:T},\delta_{1:T}:K_t=k,1\leq t\leq T} \frac{\partial}{\partial \mu^{\text{mz}}(k)} p(i_{1:T},\delta_{1:T}|\theta)$$

$$= \sum_{i_{1:T},\delta_{1:T}} \mathbf{1}_{\{K_t=k\}} \prod_{t:K_t\neq k} \phi(\delta_t,K_{t-1},i_t,i_{t-1}) \frac{\partial}{\partial \mu^{\text{mz}}(k)} \prod_{t:K_t=k} \phi(\delta_t,K_{t-1},i_t,i_{t-1})$$

$$= \sum_{i_{1:T},\delta_{1:T}} \mathbf{1}_{\{K_t=k\}} \prod_{t:K_t\neq k} \phi(\delta_t,K_{t-1},i_t,i_{t-1}) \left( \prod_{t:K_t=k} \frac{\phi(\delta_t,K_{t-1},i_t,i_{t-1})}{p(O_t^{\text{mz}}|K_t)} \right) \left( \frac{\partial}{\partial \mu^{\text{mz}}(k)} \prod_{t:K_t=k} p(O_t^{\text{mz}}|K_t) \right)$$

$$= \sum_{i_{1:T},\delta_{1:T}} \mathbf{1}_{\{K_t=k\}} \prod_t \phi(\delta_t,K_{t-1},i_t,i_{t-1}) \left( \prod_{t:K_t=k} \frac{1}{p(O_t^{\text{mz}}|K_t)} \right) \left( \frac{\partial}{\partial \mu^{\text{mz}}(k)} \prod_{t:K_t=k} p(O_t^{\text{mz}}|K_t) \right)$$

$$= \sum_{i_{1:T},\delta_{1:T}} \mathbf{1}_{\{K_t=k\}} p(s|x,\theta) \left( \prod_{t:K_t=k} \frac{1}{p(O_t^{\text{mz}}|K_t)} \right) \left( \frac{\partial}{\partial \mu^{\text{mz}}(k)} \prod_{t:K_t=k} p(O_t^{\text{mz}}|K_t) \right),$$

where

$$\frac{\partial}{\partial \mu^{\text{mz}}(k)} \prod_{t:K_t=k} p(O_t^{\text{mz}}|K_t) = \left( \prod_{t:K_t=k} p(O_t^{\text{mz}}|K_t) \right) \left( \sum_{t:K_t=k} \frac{\frac{\partial}{\partial \mu^{\text{mz}}(k)} \sum_{i_t=0}^1 p(i_t)p(O_t^{\text{mz}}|K_t,i_t)}{p(O_t^{\text{mz}}|K_t)} \right)$$

$$= \left( \prod_{t:K_t=k} p(O_t^{\text{mz}}|K_t) \right) \left( \sum_{t:K_t=k} \frac{p(i_t=0)p(O_t^{\text{mz}}|K_t,i_t=0)\frac{(O_t^{\text{mz}}-\mu^{\text{mz}}(k))}{\sigma^2}}{p(O_t^{\text{mz}}|K_t)} \right)$$

$$= \left( \prod_{t:K_t=k} p(O_t^{\text{mz}}|K_t) \right) \left( \sum_{t:K_t=k} p(i_t=0|K_t,O_t^{\text{mz}})\frac{(O_t^{\text{mz}}-\mu^{\text{mz}}(k))}{\sigma^2} \right).$$

$$\Rightarrow \frac{\partial}{\partial \mu^{\text{mz}}(k)} \log p(s|x,\theta) = \frac{1}{p(s|x,\theta)} \sum_{i_{1:T},\delta_{1:T}} \mathbf{1}_{\{K_t=k\}} p(i_{1:T},K_{1:T}|\theta) \sum_{t:K_t=k} p(i_t=0|K_t,O_t^{\text{mz}})\frac{(O_t^{\text{mz}}-\mu^{\text{mz}}(k))}{\sigma^2}$$

$$= \frac{1}{p(s|x,\theta)} \sum_{t=1}^T p(i_t,K_t=k|\theta)p(i_t=0|K_t,O_t^{\text{mz}})\frac{(O_t^{\text{mz}}-\mu^{\text{mz}}(k))}{\sigma^2}$$

$$= \sum_{t=1}^T p(i_t,K_t=k|s,\theta)p(i_t=0|K_t,O_t^{\text{mz}})\frac{(O_t^{\text{mz}}-\mu^{\text{mz}}(k))}{\sigma^2}, \tag{1}$$

where we equivalently write $p(s|x,\theta) = p(i_{1:T},\delta_{1:T}|\theta) = p(i_{1:T},K_{1:T}|\theta)$ due to the deterministic relationship $\delta_t = K_t - K_{t-1}$.

## 3 Theseus Unsupervised Learning using Coordinate Ascent

Using the model's Fisher scores, Theseus parameters $\theta$ may be learned via maximum likelihood estimation. We present an alternate learning algorithm which we compare to maximum likeli-

**Algorithm 1** Unsupervised Learning in Theseus using Coordinate Ascent
___
1: **while** not converged **do**
2:     **for** $i = 1, \ldots, n$ **do**
3:         $\hat{x}^i \leftarrow \operatorname{argmax}_{x^i \in \mathcal{P}} \log p(s^i, x^i | \theta)$
4:     **end for**
5:     $\theta \leftarrow \operatorname{argmax}_\theta \sum_{i=1}^n \log p(s^i, \hat{x}^i | \theta)$
6: **end while**
___

hood learning in [2]. Let $s^1, s^2, \ldots, s^n$ be a dataset of spectra and define $J(x^1, \ldots, x^n, \theta) = \sum_{i=1}^n \log p(s^i, x^i | \theta)$. Optimizing this objective function, Theseus' coordinate ascent learning algorithm is defined in Algorithm 1 where, rather than relying on training labels, we use max-product inference to infer the most probable PSM for each spectrum given the current iteration's parameters, then maximize the log-likelihood with respect to $\theta$ given the most likely PSMs. We now prove that Algorithm 1 converges monotonically.

**Theorem 1.** *Algorithm 1 converges monotonically to a local optimum.*

*Proof.* We need to show that the objective function $J$ is nondecreasing with each iteration of the algorithm. Denote the learned parameters at iteration $k$ of the algorithm as $\theta_k$ and define $\hat{x}_k^i = \operatorname{argmax}_{x^i \in \mathcal{P}} \log p(s^i, x^i | \theta_{k-1})$. $\theta_k = \operatorname{argmax}_\theta J(\hat{x}_k^1, \ldots, \hat{x}_k^1, \theta)$. We thus have

$$J(\hat{x}_k^1, \ldots, \hat{x}_k^n, \theta_k) \geq J(\hat{x}_k^1, \ldots, \hat{x}_k^n, \theta_{k-1})$$
$$J(\hat{x}_k^1, \ldots, \hat{x}_k^n, \theta_{k-1}) \geq J(x^1, \ldots, x^n, \theta_{k-1}), \; \forall x^1, \ldots, x^n \in \mathcal{P}$$
$$\Rightarrow J(\hat{x}_k^1, \ldots, \hat{x}_k^n, \theta_k) \geq J(\hat{x}_{k-1}^1, \ldots, \hat{x}_{k-1}^n, \theta_{k-1})$$

□

# 4   Impact of Recalibration over Standard DRIP Search

**Table 2:** Percent improvement over uncalibrated search results for the DRIP methods plotted in Figure 1, at an FDR threshold $t = 1\%$. Largest improvement highlighted in bold. Note that at this FDR threshold, Percolator post-processing using a standard set of features may result in diminished performance (Worm-3).

| Data set | DRIP | DRIP Heuristic | DRIP Fisher |
|---|---|---|---|
| Yeast-1 | 5.4 | 10.7 | **14.8** |
| Yeast-2 | 5.2 | 8.3 | **16.6** |
| Yeast-3 | 9.2 | 10.9 | **17.7** |
| Yeast-4 | 3.4 | 7.5 | **15.1** |
| Worm-1 | 10.1 | 17.4 | **20.8** |
| Worm-2 | 1.1 | 6.7 | **11.3** |
| Worm-3 | -5.1 | 7.2 | **11** |
| Worm-4 | 0.4 | 9.9 | **16** |
| Average | 3.7 | 9.8 | **15.4** |

**Figure 1:** Performance increase of DRIP search after recalibration. Methods denoted by "Percolator" are post-processed using the Percolator SVM classifier [3], otherwise the raw PSM scores of the denoted search algorithm are used for identification. "DRIP Percolator" uses the standard set of DRIP PSM features described in [1], "DRIP Percolator, Heuristic" augments the standard set with DRIP-Viterbi-path parsed PSM features described in [1], and "DRIP Percolator, Fisher" augments the Heuristic set with the gradient-based DRIP features to the standard. XCorr $p$-value and MS-GF+ use their standard set of Percolator features, described in [1]. Search accuracy plots measured by $q$-value versus number of spectra identified for yeast (*Saccharomyces cerevisiae*) and worm (*C. elegans*) datasets.