[Reviews · NeurIPS 2017]

Reviewer 1



This paper introduces Theseus, an algorithm for matching MS/MS spectra to peptide in a D.B. This is a challenging and important task. It is important because MS/MS is currently practically the only common high-throughput method to identify which proteins are present in a sample. It is challenging because the data is analog (intensity vs. m/z graphs) and extremely noisy. Specifically, insertions, deletions, and variations due to natural losses (e.g. carbon monoxide) can occur compared to a theoretical peptide spectra. This work builds upon an impressive body of work that has been dedicated to this problem. Specifically, the recently developed DRIP (Halloran et al 2014, 2016) is a DBN, i.e. a generative model that tries to match the observed spectra to a given one in the D.B. modeling insertions/deletions etc. and computes for it a likelihood and the most likely assignment using a viterbi algorithm. A big issue in the field is calibration of scores (in this case - likelihoods for possible peptide matches from the D.B to an observed spectra) as these vary greatly between spectra (e.g. different number of peaks to match) so estimating which score is “good enough” (e.g. in terms of overall false positive rate) is challenging. Another previous algorithm, Percolator, has been developed to address this by training an SVM to discriminate between “good” matching peptides scores and “negative” scores from decoy peptides. A recent work (Halloran et al 2016) tried to then map the DRIP viterbi path to a fixed sized vector so Percolator could be applied atop of DRIP. Here, the authors propose an alternative to the above heuristic mapping. Instead, they employ Fisher Kernels where two samples (in our case - these would be the viterbi best path assigned to two candidates matches) are compared by their derivative in the feature space of the model. This elegant solution to apply discriminative models (here Percolator) to generative models (here DRIP) was suggested for other problems in the comp bio domain by Jaakkola and Haussler 1998. They derive the likelihood and its derivatives and the needed update algorithms for this, and show employing this to several datasets improve upon state of the art results. Pros: This work tackles an important problem. It results in what seems to be a significant improvement upon state of the art. It offers a new model (Theseus) and an elegant application Jaakkola and Haussler’s Fisher Kernel idea to the problem at hand. Overall, this appears to be high-quality computational work advancing an important topic. Cons: Major points of criticism: 1. Writing: While the paper starts well (intro) it quickly becomes impenetrable. It is very hard to follow the terms, the variables and how they all come together. For example, Fig1 is referred to in line 66 and includes many terms not explained at that point. Sec. 2.1 refers to c_b which is not defined (is it like c(x) ?) How are the natural losses modeled in this framework, Lines 137-145 are cryptic, evaluation procedure is too condensed (lines 221-227). In general, the authors spend too much space deriving all the equations in the main text instead of actually explaining the model and what is new compared to previous work. This makes evaluating both the experiments and the novelty hard. To make things worse, the caption of the main figure (Fig3) refers to labels/algorithms (“Percolator”, “DRIP Percolator Heuristic”) which do not appear in the figure. Given the complexity of the domain and the short format of NIPS papers, we recommend the authors revise the text to make the overall model (Theseus) and novel elements more clear, and defer some of the technical details to a longer more detailed supplementary (which in this case seems mandatory).

Reviewer 2



Summary: Tandem mass spectrometry (MS/MS) is used to identify the proteins in a complex biological sample. In the past, discriminative classifiers were used to identify the top ranked peptides (subsequence of protein). Many heuristically derived features were shown to improve the performance. In this work, the author proposed a novel unsupervised generative model "Theseus" -- a Dynamic Bayesian Model with Fisher kernel. The author claim that providing the gradient-based features from Theseus as the input features to the discriminative models could substantially improve the prediction quality. Supportive real world experiments were provided to show that the proposed method leads the state-of-art identification accuracy. Qualitative Evaluation: Quality: The author supported the proposed method with experiments on several real world datasets. The method is compared to many previous approaches and shows significant improvements. However, the paper lack of some discussion about the computational efficiency: how much overtime is needed when using proposed method to extract features? Clarity: The paper is well organized. The experiment details are well described. However, when introducing the dynamic bayesian model, it might be better if the author could better introduce the variables and symbols. For example, the author introduced spectrum s in section 2 line 68. But its definition was not introduced until section 3 in page 4. Also, line 130, O^{mz} and O^{in} is not well-defined, I think the (O^{mz},O^{in}) pair means a particular peak's m/z and intensity? When reading the paper with so many symbols, concepts and overloading functions, I think it might be helpful if the author could provide a table listing the symbols and their meanings. Some minor issues: In page 4, line 130, "x a candidate peptide of length m" should be "x be a candidate peptide of length m" In page 4, line 135, "specrum" -> "spectrum" Originality: This work provides a novel unsupervised method on a traditional computational task. It follows the trend of generating better features to discriminative models to improve the accuracy. Relevances and comparisons to previous methods are well discussed in the paper. Significance: The works provides a new way to extract features based on unsupervised learning requires no heuristic features for protein identification in tandem mass spectrum. The experiment results shows substantial increase in prediction accuracy comparing to previous work and leads the new state-of-art.